

# Accounting for non-rainfall moisture and temperature improves litter decay model performance in a fog-dominated dryland system

J. Robert Logan[1,2], Kathe E. Todd-Brown[3], Kathryn M. Jacobson[4], Peter J. Jacobson[4], Roland Vogt[5], Sarah E. Evans[1,2]

[1]W. K. Kellogg Biological Station, Hickory Corners, MI, USA
[2]Department of Integrative Biology, Michigan State University, East Lansing, MI, USA
[3]Department of Environmental Engineering Sciences, University of Florida, Gainesville, FL, USA
[4]Department of Biology, Grinnell College, Grinnell, IA, USA
[5]Institute of Meteorology, Climatology and Remote Sensing, University of Basel, Basel, Switzerland

*Correspondence to*: J. Robert Logan (loganja3@msu.edu)

**Abstract.** Historically, ecosystem models have treated rainfall as the primary moisture source driving litter decomposition. In many arid and semi-arid lands however, non-rainfall moisture (fog, dew, and water vapor) plays a more important role in supporting microbial activity and carbon turnover. To date though, we lack a robust approach for modeling the role of non-

rainfall moisture in litter decomposition. We developed a series of simple litter decay models with different moisture sensitivity and temperature sensitivity functions to explicitly represent the role of non-rainfall moisture in the litter decay process. To evaluate model performance, we conducted a 30-month litter decomposition study at six sites along a fog/dew gradient in the Namib Desert, spanning almost an eight-fold difference in non-rainfall moisture frequency. Litter decay rates in the field correlated with fog and dew frequency but not with rainfall. Including either temperature or non-rainfall moisture

sensitivity functions improved model performance, but the combination of temperature and moisture sensitivity together provided more realistic estimates of litter decomposition than relying on either alone. Model performance was similar regardless of whether we used continuous moisture sensitivity functions based on relative humidity or a simple binary function based on the presence of moisture, though a Gaussian temperature sensitivity outperformed a monotonically increasing $Q_{10}$ temperature function. We demonstrate that explicitly modeling non-rainfall moisture and temperature together is necessary to

accurately capture litter decay dynamics in a fog-affected dryland system and provide suggestions for how to incorporate non-rainfall moisture into existing Earth system models.

## 1 Introduction

Drylands play an important part in the global carbon cycle, but we still lack a strong understanding of carbon cycling in these systems. Historically, ecosystem models have underestimated dryland litter decomposition rates (Parton et al., 2007;

Adair et al., 2008). This is partly because the models are driven by rainfall, assuming little to no decay between precipitation events. While rainfall pulses play a large role in dryland systems (Noy-Meir, 1973; Seely and Louw, 1980), considering rain





alone does not fully capture litter decomposition in these systems. This may be partially because much decomposition occurs at and above the soil surface, and aboveground litter decomposition is less sensitive to large rain pulses than is belowground decay (Austin, 2011; Jacobson and Jacobson, 1998). Abiotic processes including photodegradation, aeolian erosion, and

thermal degradation that drive aboveground litter decomposition can degrade litter regardless of moisture conditions (Austin, 2011) and rain events as little as 1 mm can facilitate microbial activity (Collins et al., 2008). Finally, non-rainfall moisture (NRM: fog, dew, and water vapor) can support substantial biotic decomposition of plant litter, even in the absence of rain (Jacobson et al., 2015; Dirks et al., 2010; Wang et al., 2017; Logan et al., 2021). These findings demonstrate that carbon and nutrient cycling in drylands are not restricted to precipitation pulses and that NRM is a crucial driver of dryland biogeochemical

cycles. As our understanding of the importance of NRM in arid and semi-arid ecosystems evolves, we need to update our conceptual and predictive models to incorporate these important drivers of ecosystem processes.

Despite growing recognition of NRM's importance, current litter decay models do not explicitly account for its ability to support decomposition. This is partly because field-based studies of NRM-driven decomposition are scarce and so far, have mostly focused on documenting single cases and understanding mechanisms. Recent studies have shown that the rate of NRM-

driven decomposition depends on many factors including the frequency of humid conditions (Evans et al., 2020), the composition of decomposer communities (Logan et al., 2021; Wenndt et al., 2021), and interactions with other processes like photodegradation (Wang et al., 2015; Gliksman et al., 2017; Logan et al., 2022). These insights have been very helpful in demonstrating that NRM-driven decomposition occurs and identifying its various mechanisms. However, before we can incorporate NRM into mechanistic Earth system models we need multi-year studies that quantify the relationship between

NRM and mass loss across a range of environmental conditions (Bonan et al., 2013), something that has not been done to date.

One recent attempt to model NRM-driven decomposition has shed light on this challenge. Evans et al. (2020) developed a model that treated decomposition as a pulse process that could be triggered by either rain or NRM when conditions met a given criterion (i.e., when relative humidity was above a given threshold or when dew was present as determined by a leaf wetness sensor). They found that accounting for NRM produced mass loss estimates that were considerably higher than

those from a rain-only model and that these new estimates were within the range observed in the field. This approach showed that NRM can improve mass loss estimates, but it included several simplifying assumptions that need to be tested before NRM can be incorporated into models more generally. First, the authors modelled annual mass loss by measuring instantaneous respiration rates and scaling them up to annual timescales. This showed that the NRM-driven biotic activity on the scale of individual events can be used to estimate long-term mass loss rates over several months, albeit with wide error estimates. A

better approach would be to validate model predictions by formally integrating rates of mass loss at multiple sites and in multi-year field studies (Bonan et al., 2013). Studies where NRM meteorology and decomposition are both measured and quantifiably linked to one another are currently lacking.

Second, their model treated decomposition as essentially a pulse process that could be triggered by either rainfall or NRM, but responded similarly to both (in other words, as long as the threshold condition was met, decomposition was

considered to be "on"). While rainfall and NRM may induce similar decomposition rates for a similar moisture level, this



approach does not allow the possibility of continuous responses. For example, $CO_2$ fluxes are strongly correlated with litter moisture content (Jacobson et al., 2015), which varies with relative humidity (Tschinkel, 1973; Dirks et al., 2010), so a sensitivity function that allows instantaneous decay rates to vary depending on the magnitude of the NRM event may be more appropriate than a simple threshold trigger. Finally, their model did not include temperature dependence, despite

decomposition being highly sensitive to temperature in almost all terrestrial systems (Sierra, 2012; Sierra et al., 2015). Relative humidity is closely linked to air temperature, and average temperature during NRM events is often considerably lower than during rain events (Logan et al., 2021). Developing more powerful NRM-driven litter decay models may therefore require incorporating continuous moisture responses and temperature sensitivities to accurately capture decomposition dynamics, though to date these remain untested.

We set out to determine whether incorporating NRM into a simple litter decay model improved model performance in an NRM-dominated system. We tested multiple potential relationships between meteorological variables and litter decay rates in an attempt to parameterize a model of NRM-driven decomposition. We had two main objectives:

1. Use a novel dataset to evaluate multiple methods of modeling litter decomposition as a function of NRM.
2. Determine how important temperature sensitivity is in NRM-driven litter decomposition models.

Since existing studies examining decomposition to different moisture regimes are limiting (Jacobson et al., 2015; Evans et al., 2020), we draw upon literature on soil organic matter decomposition and rainfall-driven litter decomposition to identify potential moisture and temperature sensitivity functions (Sierra et al., 2015). To evaluate models, we conducted a 30-month, multi-site litter decomposition study that spanned an eight-fold magnitude of NRM frequency. By placing litter across this gradient and making continuous meteorological measurements alongside mass loss, we were able to quantify the

relationship between NRM and litter decay on a multi-year timescale for the first time. Finally, we used a Bayesian model-data integration approach to parameterize mass loss models using several temperature and moisture sensitivity functions and used model selection criteria to identify the best models.

## 2 Methods

### 2.1 Empirical measurements

We conducted our study in the central Namib Desert in western Namibia. The Namib Desert is a coastal fog desert, with a steep NRM gradient across a narrow geographic range (Eckardt et al., 2013). Rain is scarce in the Namib and NRM is expected to be responsible for the vast majority of litter decomposition (Evans et al., 2020). We leveraged the FogNet weather array, a network of meteorological stations throughout the central Namib Desert that is part of the Southern African Science Service Centre for Climate Change and Adaptive Land Management (SASSCAL; www.sasscalweathernet.org) and

maintained by the Gobabeb Namib Research Institute (www.gobabeb.org) (Fig. A1). Each station measures air temperature, relative humidity, wind speed and direction, soil temperature, leaf wetness state, rainfall, and fog precipitation on a Juvik fog screen. The sites are all located within 70 km of one another but span an order of magnitude in NRM frequency: wet conditions



(fog or dew) occur for 3.1% of the period (quantified by hours wet) at the driest site and 25.3% at the wettest; a full characterization of meteorology across these sites was part of this study. Weather data were recorded once per minute and converted to hourly averages for analysis.

At six sites, we deployed senesced tillers of *Stipagrostis sabulicola* to monitor mass loss. *S. sabulicola* is the dominant grass in the Namib Sand Sea with widely distributed congenerics across Africa and Asia (Roth-Nebelsick et al., 2012; GrassBase - The Online World Grass Flora., 2021). Since litter-associated fungal communities can respond differently to NRM based on their history of exposure to different moisture regimes (Logan et al., 2021), we collected all tillers from the same site (Gobabeb) so the initial fungal community would be the same. To avoid potential microclimate effects of traditional litter bags (Xie, 2020), we measured mass loss by placing tillers in litter racks, custom-made wooden frames designed to hold grass tillers while keeping them completely exposed to ambient solar radiation and moisture conditions (Fig. A2) (Evans et al., 2020; Logan et al., 2021). Every six months for 30 months (19 January 2018 to 12 August 2020; 936 days in total), we collected a subset of ten tillers at each site and weighed them. Tillers were destructively harvested at each time point; so in our final dataset, each tiller was weighed prior to deployment and once again when it was collected.

To assess the effects of NRM on litter decomposition throughout the decay process, we deployed litter at two stages of decay. Categories were based on previous observations of *S. sabulicola* decay *in situ* in the Namib (Logan et al., 2021). Early-stage tillers were tillers that had senesced in the preceding two months, had no visible fungal growth, and had visibly intact cuticles (Fig. A2). Late-stage tillers were harvested from upright plants that had likely been standing for at least one year post-senescence and were characterized by coverings of light and dark-pigmented fungi and a cracked cuticle that was considerably more permeable to water. Previous work found similar measures of gross litter quality (including C:N ratios, total lignin, and lignin:N content) between litter at these two stages, and found that the primary difference between the two is the level of fungal colonization and state of cuticle degradation, with late-stage tillers harboring much larger fungal communities (Logan et al., 2021). Since we only collected standing grass litter that had not fallen over yet, our terminology of "early" and "late" does not reflect the entire decomposition process but is meant to highlight relative successional differences between the litter stages based on time since senescence and saprophytic community size.

## 2.2 Model description

To model the effect of NRM on litter decomposition, we began by modeling decay rates using a simple exponential model of the form:

$$M(t) = M_0\, e^{-k_{eff}\, t} \tag{1}$$

Where $M(t)$ is mass at time $t$, $M_0$ is initial mass, and $k_{eff}$ is the effective litter decay constant. This approach captures typical litter decay dynamics, with a rapid initial decay phase followed by slower mass loss over time, but does not differentiate between slow and rapid litter pools. We determined an effective decay rate for each site and litter stage, plotting this as a function of the total NRM time and accumulated rainfall at that site.



This approach, whereby we fit a separate effective decay rate for sites with different climates, is a common approach to describe how litter decomposition varies under different climatic conditions (Zhang et al., 2008). However, because it treats mass loss as solely dependent on the decay rate and time, this approach does not explicitly include temperature or moisture. To determine how moisture and temperature influenced litter decay, we incorporated NRM and temperature dependence by allowing them to modify an intrinsic litter decay ($k_{int}$) term, which represents the rate of litter decay under ideal, non-limiting

conditions according to:

$$\frac{dM(t)}{dt} = -k_{int} \, g(t) \, h(t) \, M(t) \tag{2}$$

Or equivalently,

$$M(t) = M_0 \, e^{-k_{int} \, t \, g(t) \, h(t)} \tag{3}$$

Where h(t) and g(t) are sensitivity functions for NRM and temperature respectively. Unlike the simple model

described by Equation 1, in this model, the litter decay rate ($k_{int}$) is the maximum rate under ideal temperature and moisture conditions, which is then modified downward by the sensitivity functions (with the exception of $Q_{10}$ temperature sensitivity function that allows increasing decomposition above a reference temperature); see next section for sensitivity functions. This allowed us to test specific hypothesized relationships between moisture and litter decay rates, both within and between sites depending on how we choose to fit the parameters (i.e. separate fits for each site or global parameter estimates). Using a one-

pool model allowed us to simplify the intrinsic decay component of the model and focus on the effect of different temperature and moisture sensitivities. We discretized the model using hourly meteorological data, calculating the rate of mass loss for each hour as:

$$\frac{M_{n+1}}{M_n} = 1 - \Delta t_n \, k_{int} \, g(t_n) \, h(t_n) \tag{4}$$

### 2.3 Sensitivity functions

Since litter decomposition can occur in response to dew and fog (Jacobson et al., 2015) or water vapor under humid conditions even in the absence of liquid water (Dirks et al., 2010), we tested separate sensitivity functions based on either relative humidity levels, or based on a measurement of the presence of liquid water. Sensitivity functions are presented in Table 1 and shown in Fig 1. The threshold model is binary, allowing decomposition to happen at the intrinsic litter decay rate if and only if relative humidity is above a specified threshold ($R_T$). This simple approach has yielded mass loss estimates

similar to those measured in the field previously (Evans et al., 2020). To account for possible saturation at high relative humidities, we also evaluated a logistic sensitivity model that allows the rate of decomposition potential to slow as relative humidity approaches 100%. The exponential moisture model allows decomposition rates to increase exponentially with relative humidity, reflecting the relationship between litter moisture content and relative humidity that is often seen in both controlled (Tschinkel, 1973) and field conditions (Dirks et al., 2010). Each moisture sensitivity function was normalized to 1

when relative humidity was 100%. Finally, we tested a fourth function based on the presence or absence of moisture as measured by a leaf wetness sensor, in which decomposition occurred at the intrinsic decay rate when the wetness sensors were





wet and not at all when conditions were dry. Previous work showed that relative humidity can accurately predict leaf wetness state (Sentelhas et al., 2008; Evans et al., 2020), so we expected this model to perform similarly to the threshold model.

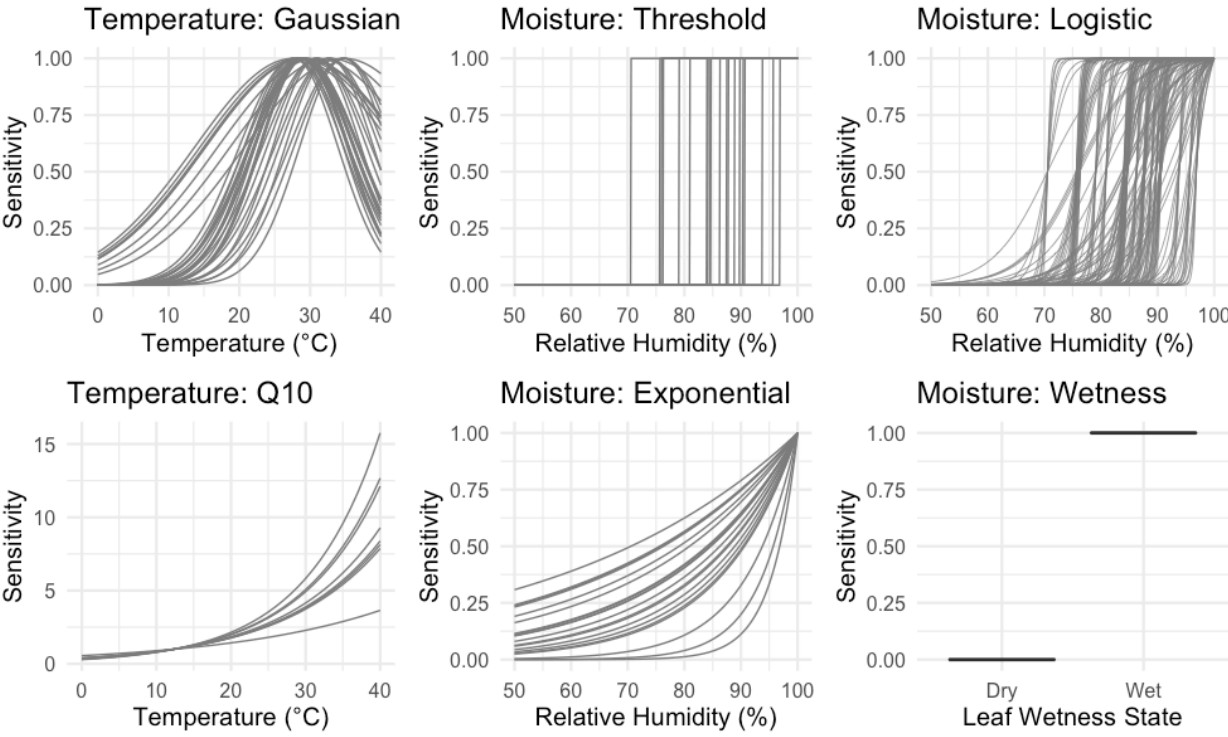

**Figure 1: Temperature and NRM sensitivity functions included in the models. Each curve shows one parameter combination chosen by randomly sampling using a normal distribution around a specified set of priors as identified in Table 2 (n = 895,230 total combinations). The wetness moisture function has no parameter and is simply the proportion of time during each hour that the leaf wetness sensor detected the presence of moisture.**

To model temperature dependence, we tested two common temperature sensitivity functions: a $Q_{10}$ model and a Gaussian distribution. $Q_{10}$ sensitivity is a monotonically increasing function that is used to model many biological process including litter decomposition (Sierra et al., 2015). Each increase of 10°C above a reference temperature ($T_{ref}$, often the site's mean temperature), results in an acceleration of the process in question by a given amount, called the $Q_{10}$ coefficient. To account for possible negative temperature dependence above an optimum temperature ($T_{opt}$), we also tested a Gaussian temperature sensitivity function. A Gaussian function is often particularly well suited for describing aggregated responses of entire communities (Low-Décarie et al., 2017), as is the case for the fungal communities on our tillers (Logan et al., 2021). Temperature sensitivity was normalized to 1 at $T_{opt}$ in the Gaussian model and $T_{ref}$ in the $Q_{10}$ model. We tested each combination of moisture and temperature functions (as well as moisture-only and temperature-only versions) for a total of 15 different model structures and 895,230 model-parameter combinations.






**Table 1. Moisture and temperature sensitivity functions. The first three moisture functions are based on relative humidity and the fourth is based on leaf wetness state. Moisture functions are normalized to 1 at 100% relative** 185 **humidity and temperature sensitivity functions are normalized to 1 at $T_{ref}$ and $T_{opt}$.**

| Class | Name | Model | Parameters |
|---|---|---|---|
| Moisture | Threshold | $h(RH) = if(RH > R_T)$ | $R_T$ = relative humidity threshold |
| Moisture | Exponential | $h(RH) = 2^{\frac{100-RH}{R_{0.5}-100}}$ | $R_{0.5}$ = RH value at half saturation point |
| Moisture | Logistic | $h(RH) = \frac{1 + e^{r\,(R_{0.5}-100)}}{1 + e^{r\,(R_{0.5}-RH)}}$ | r = logistic growth rate<br>$R_{0.5}$ = RH value at half saturation point |
| Moisture | Wetness | $h(LWS) = LeafWetnessState$ | None |
| Temp. | $Q_{10}$ Model | $g(T) = Q_{10}^{(T-T_{ref})/10}$ | $Q_{10}$ = $Q_{10}$ coefficient<br>$T_{ref}$ = Reference temperature |
| Temp. | Gaussian | $g(T) = e^{-0.5\left(\frac{T-T_{opt}}{stDev}\right)^2}$ | stDev = standard deviation<br>$T_{opt}$ = Optimal temperature |

To understand the nature of the different models and compare them across a range of conditions, we performed two model runs. First, we explored a large parameter space to determine how parameters interact with one another across a wide range of hypothetical conditions. This included parameter values outside of realistic ranges (for example, relative humidity 190 thresholds from 5-99% and an intrinsic litter turnover time from 0.1-100 years). This allowed us to see how parameters interacted with each other within the different models and explore general properties of each model. Next, to assess which models performed best under realistic conditions, we constrained the parameter space to more accurately reflect real world parameter values. For this model run, we determined optimal values for each parameter based on lab and field incubations and then randomly varied parameter combinations around these values; see next section for details. Parameter definitions as 195 well as constrained values used in the second model run are reported in Table 2. Figure 1 shows the range of temperature and moisture sensitivities we used in the constrained model run.

We used the Akaike information criterion (AIC) to compare the constrained models to one another to determine which was the best fit to the data. AIC is a model selection criterion that rewards goodness of fit based on a log likelihood function while penalizing models with greater parameters to reduce overfitting biases (Aho et al., 2014). We report AIC values for all 200 combinations of models from the constrained parameter run to compare model performance under realistic scenarios.



**Table 2. Parameter definitions and values used to constrain the second model run to realistic conditions. Values were randomly varied around means and standard deviations shown, with "n" denoting the number of iterations used for each parameter (n = 895,230 total model-parameter combinations). For $Q_{10}$, $T_{opt}$, and stDev, models were run with two**
**standard deviations (i.e. twice the value shown below).**

| Parameter | Definition | Model (type) | Value | Justification |
|---|---|---|---|---|
| $Log_{10}$ Turnover Time (1/k) | Intrinsic turnover time (i.e. turnover time under ideal temp & moisture conditions) | All | $1 \pm 1$ year (n = 20) | Estimated from maximum respiration rate from previous studies |
| $T_{ref}$ | Reference temp for $Q_{10}$ function | $Q_{10}$ (temp) | 12.3°C | Mean temp when wet |
| $Q_{10}$ | $Q_{10}$ sensitivity | $Q_{10}$ (temp) | $2.38 \pm 0.292$ °C (n = 8) | Temperature incubations (Fig 2) |
| $T_{opt}$ | Optimum temperature for Gaussian distribution | Gaussian (temp) | $29.7 \pm 2.37$ °C (n = 8) | Temperature incubations (Fig 2) |
| stDev | SD around $T_{opt}$ for Gaussian distribution | Gaussian (temp) | $6.59 \pm 3.02$°C (n = 4) | Temperature incubations (Fig 2) |
| $R_{0.5}$ | RH value where moisture sensitivity is 50% of maximum | Exponential (NRM) Logistic (NRM) | $90 \pm 10\%$ (n = 20) | Range of humidity conditions during which NRM typically occurs (Evans et al. 2020) |
| $R_T$ | RH value above which decomp is "on" | Simple threshold (NRM) | $90 \pm 10\%$ (n = 20) | Range of humidity conditions during which NRM typically occurs (Evans et al. 2020) |
| r | Rate of logistic growth | Logistic (NRM) | $1 \pm 1$ (n = 8) | Smaller values approximated a straight line; higher values resembled the simple threshold model |

## 2.4 Constraining parameter space

To constrain temperature parameters, we performed a lab incubation of *S. sabulicola* tillers. We varied the
temperature from 10-35°C at 5°C steps, allowing litter to equilibrate for 60 minutes before measuring respiration. We sprayed eight tillers with sterile deionized water until they were saturated to stimulate fungal activity and placed them in 55 ml acrylic tubes connected to a LI-8100A gas analyzer (LI-COR Biosciences, Lincoln, Nebraska, USA), measuring mean flux during 3 minute incubations. To measure the response of the specific fungal communities associated with litter used in the field study,



all tillers used in the lab incubation were collected at Gobabeb, the same site where litter in the mass loss experiment was
collected.

To calculate $Q_{10}$, we excluded the measurements at 35°C (when response becomes negative) and then used the 'Q10'
function in the *respirometry* package in R to calculate a separate $Q_{10}$ value for each tiller (Birk, 2021; R Core Team, 2020).
For the reference temperature, we used the mean temperature when leaf wetness sensors were "wet" across all sites (12.3°C).
This value was fairly constant among sites, varying by less than 0.9°C (Fig. A3). For the Gaussian temperature sensitivity
parameters, we used the 'optim' function in R to find the optimum temperature ($T_{opt}$) and standard deviation (stDev) around
the optimum after normalizing flux rates to the maximum rate measured for each tiller (R Core Team, 2020).

The turnover time represents the litter's intrinsic decay rate under ideal temperature and moisture conditions and is
equivalent to the inverse of $k_{int}$, the exponential parameter in the decay function. To place a lower boundary on this value, we
examined previous studies that measured respiration from *Stipagrostis sabulicola* under wet conditions and extrapolated to
estimate a minimum turnover time (in years) under ideal, non-limiting conditions. Jacobson et al. (2015) reported respiration
rates from wet *S. sabulicola* tillers as high as 1.5 μg $CO_2$-C g$^{-1}$ litter hr$^{-1}$, corresponding to an intrinsic turnover time of ~0.63
years, assuming 50% of plant litter mass is carbon. This is within the range of intrinsic turnover rates reported for other grasses
(Zhang et al., 2008). We therefore used a turnover time with a mean of 1 year around a log-normal distribution. We varied
the logistic growth parameter (r) of the logistic moisture sensitivity function around a value 1, because at much lower values,
it began to resemble a straight line (i.e. no longer logistic sensitivity) and at higher values, it converged on the simple threshold
model.

## 3 Results

### 3.1 Meteorological conditions and temperature incubations

Moisture conditions varied substantially among the sites. Duration of wetness during the study period (as determined
by leaf wetness sensors) ranged from 672 hours (3.1% of total hours) at the driest site (Garnet Koppie) to 5672 hours (25.3%
of total hours) at the wettest site (Kleinberg). Drier sites tended to be warmer; mean temperature when dry was 2.3°C warmer
at the warmest site (Garnet Koppie) than at the coolest site (Kleinberg) (Table 3). Temperatures during NRM events were
lower and less variable than temperatures during dry periods (Table 3). Wet conditions almost never occurred when
temperatures were above 20°C at any site (Fig. A3). Average relative humidity differed among the sites and was correlated
with hours of wetness. Rainfall occurred at all sites during the study period, ranging from 26.4–64.2 mm, but did not correlate
with NRM frequency. The optimum temperature for respiration in the incubations was 30°C, with flux rate dropping at 35°C
(Fig 2).





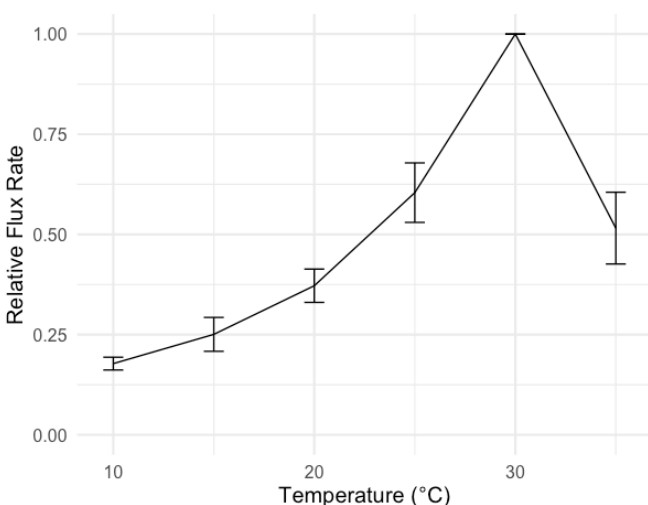

**Figure 2: Temperature sensitivity of respiration from *S. sabulicola* tillers in a lab incubation, used to constrain temperature parameters (mean ± 1 S.E.M., n = 8). Flux rate is normalized to the rate at 30°C. Tillers were sprayed with deionized water until saturated and respiration was measured at 5°C intervals.**

**Table 3. Summary of meteorological conditions at each site during the study showing mean temperature when dry, mean temperature when wet, wet hours during the entire study period (as determined by leaf wetness sensors), the proportion of total time when conditions were wet, accumulated rainfall during the study period, and mean relative humidity throughout the study period. Temperature ranges in parentheses report the middle 95% of data. Mean temperatures apply to the time period used in this study but should not be used to infer mean annual temperatures for each site since the study lasted 2.5 years and therefore data from Jan-Aug is represented more than Sep-Dec. Full names for sites are included in Fig. A1.**

| Site | $Temp_{dry}$ (°C) | $Temp_{wet}$ (°C) | Wet Hours | Prop. Time Wet | Rain (mm) | Mean Rel. Humidity (%) |
|------|-------------------|-------------------|-----------|----------------|-----------|------------------------|
| GK | 22.4 (12.67 - 32.34) | 12.2 (6.88 - 19.17) | 674 | 3.0% | 61.8 | 37.5 |
| GB | 22.2 (11.35 - 33.42) | 12.4 (6.24 - 17.91) | 1645 | 7.3% | 64.2 | 44.8 |
| S8 | 21.4 (10.92 - 32.49) | 11.7 (5.66 - 17.4) | 1930 | 8.6% | 26.4 | 46.9 |
| VF | 21.7 (11.75 - 32.37) | 12 (6.34 - 16.38) | 2214 | 9.9% | 33.7 | 50.2 |
| MK | 22 (12.73 - 32.09) | 12.6 (8.00 - 16.59) | 2810 | 12.5% | 44.5 | 53.6 |
| KB | 20.1 (11.01 - 30.91) | 12.4 (6.69 - 17.07) | 5672 | 25.3% | 56.6 | 67.7 |





### 3.2 Litter mass loss

In general, mass loss was greater at sites with more non-rainfall moisture and lower at sites with less NRM (Fig 3). There was a significant three-way interaction between litter stage, site, and time (Table S4.1). Within each site, early-stage and late-stage litter decomposed at comparable rates for the first 18 months, but diverged after that depending on the site (Fig 3). After 24 months at the two driest sites, early-stage litter lost more mass than did late-stage litter. At the four wettest sites however, late-stage litter experienced the greater mass loss (Fig 3).

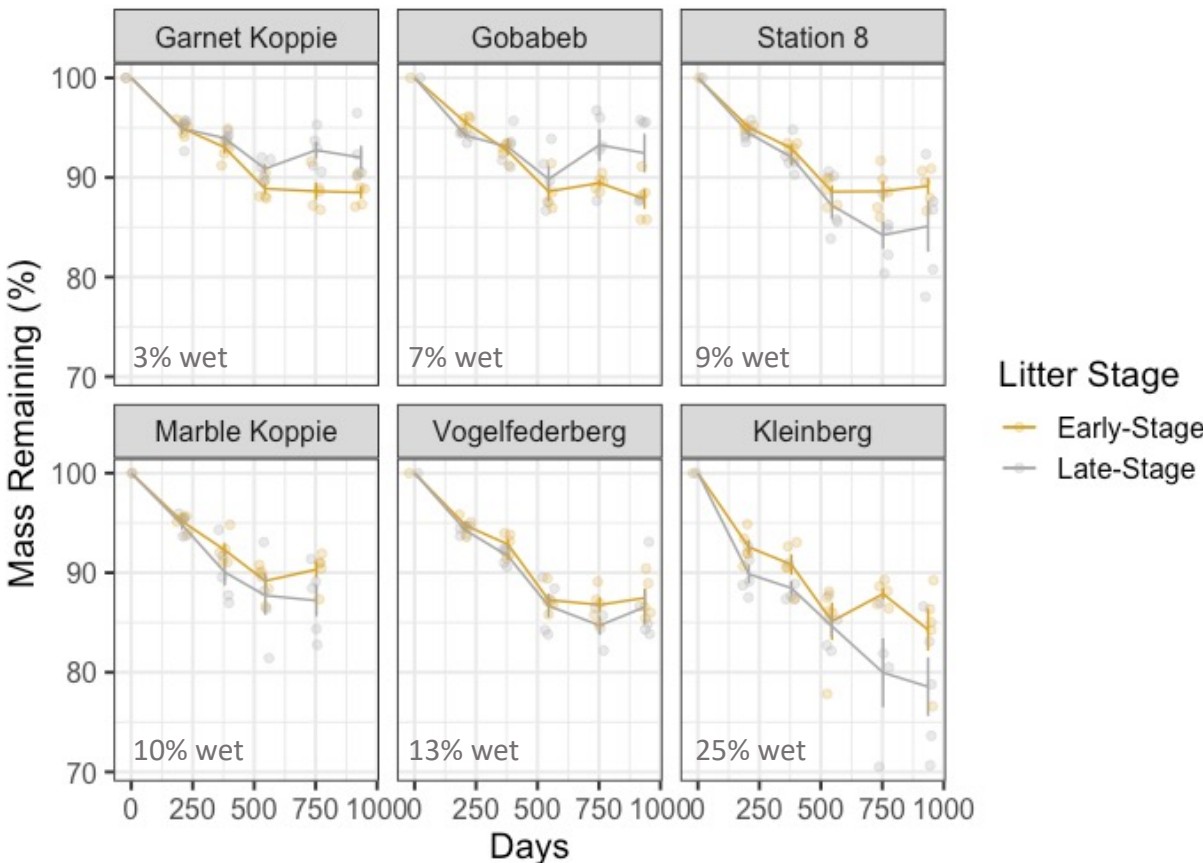

**Figure 3: Mass loss for early-stage (yellow) and late-stage (grey) tillers at each site (mean ± 1 S.E.M.). Percentage values in the bottom of each panel show the average proportion of time throughout the study period that the site has liquid water, as determined by a leaf wetness sensor. Note: tillers did not actually increase in mass; the apparent increase at some time points in some panels is merely a result of variation among tillers, since we destructively harvested tillers at each time point and could therefore not take repeated measurements of each tiller.**

When we used a simple exponential decay model without temperature and moisture sensitivity (Equation 1), the effective decay rate at each site was correlated with NRM duration but not with accumulated rainfall (Fig 4). Late-stage litter





(i.e., tillers with more well-established fungal communities) responded more strongly to NRM than did early-stage litter; for every additional 1000 hours of wetness at a site, effective decay rate increased 0.0043 for early-stage litter and 0.014 for late-stage litter (Fig 4).

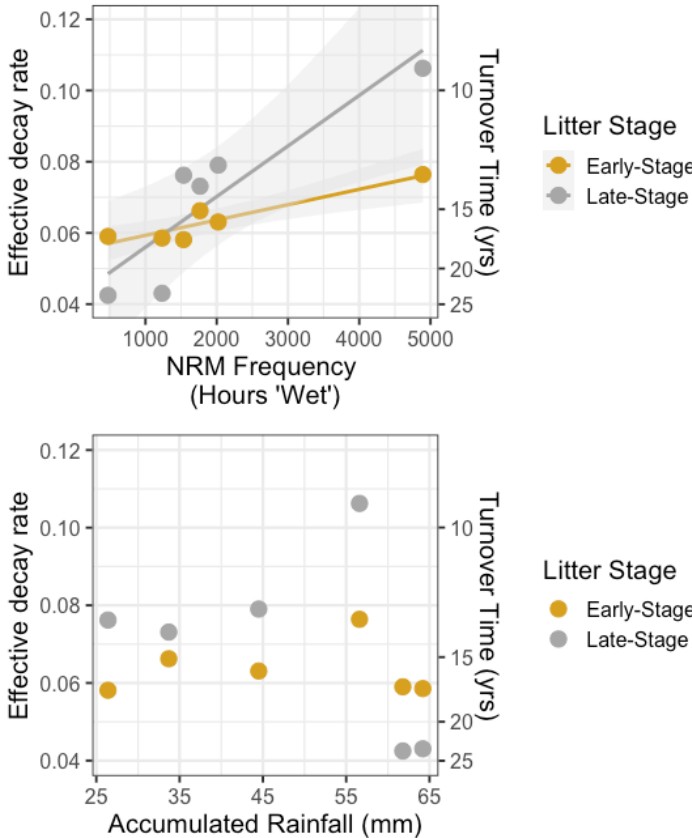


**Figure 4: Effective decay rate calculated without explicit temperature or NRM sensitivity (Equation 1) relative to NRM frequency and accumulated rainfall during the study period. Among sites, decay rate constant was strongly correlated with the proportion of time that a site experienced NRM conditions (Early-Stage: $R^2 = 0.87$, P = 0.007, slope = $4.311*10^{-6}$; Late-Stage: $R^2 = 0.80$, P = 0.02, slope = $1.421*10^{-5}$) but was uncorrelated with total rainfall (Early-Stage: $R^2 = 0.01$,**
**P = 0.87; Late-Stage: $R^2 = 0.14$, P = 0.46).**

**3.3 Model parameter space exploration**

For the three NRM sensitivity functions based on relative humidity, parameter values showed a tradeoff between turnover time and RH thresholds (Fig 5): parameter combinations with the lowest AIC scores featured either slow turnover
times and a low RH threshold (bottom right of plots) or faster turnover times and high RH thresholds (upper middle of plots). When we fit parameters separately for each site instead of globally, AIC values improved, but the actual values of the best





parameter combinations did not change (Fig. A4). Similarly, fitting parameters separately to early- and late-stage tillers did not produce different optimal parameter values (Fig. A5).

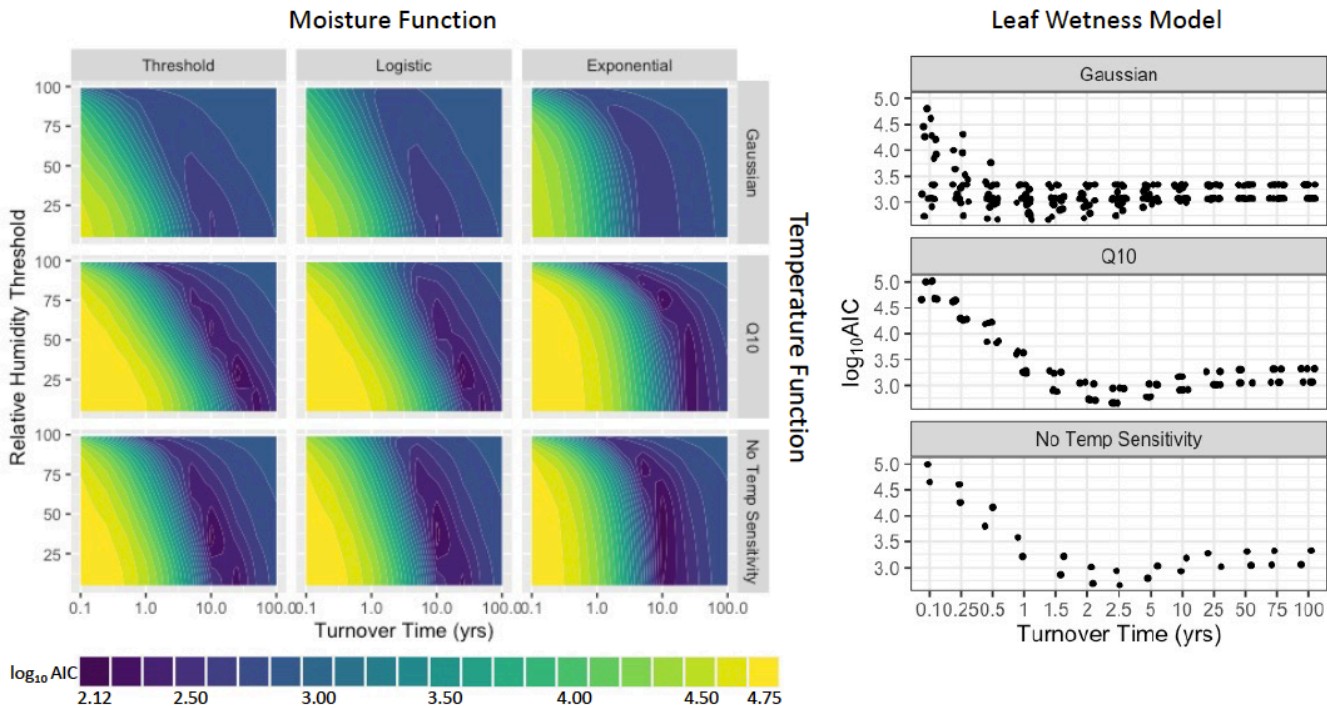


**Figure 5: Parameter fits for the first model run showing parameter combinations across a wide range of hypothetical conditions. (Left) The three humidity-based NRM functions showing the relationship between turnover time (1/$k_{int}$) and relative humidity threshold ($R_T$ or $R_{0.5}$). Colors represent log$_{10}$AIC scores. (Right) Parameter estimation for the leaf wetness-based moisture function showing log$_{10}$AIC as a function of turnover time (1/$k_{int}$). Plots have different**
**numbers of points because of different numbers of temperature parameters that were tested (the Gaussian temperature function has two, the Q$_{10}$ function has one, and the bottom plot has no temperature parameters, only early- and late-stage tiller combinations).**

Models that included Q$_{10}$ temperature sensitivity converged on slower intrinsic decay rates (i.e., longer turnover
times) than did those using a Gaussian temperature sensitivity or temperature-independent models (Fig 5). The wetness sensitivity functions yielded an optimal litter turnover time of 2.5 years under a moisture-only and Q$_{10}$ temperature sensitivity model (Fig 5). Using a Gaussian temperature sensitivity yielded a faster intrinsic decay with an optimal turnover time from 0.5-1.5 years.





### 3.4 Model performance comparison

Models that included NRM sensitivity had better fits than did the simple litter decay model, but the best models included both NRM and temperature sensitivity (Fig 6). While model fit improved (AIC scores were lower) whenever NRM sensitivity was included, the degree to which NRM sensitivity improved the model fit depended on the temperature sensitivity function. In particular, models with Gaussian temperature sensitivity performed better than did those with $Q_{10}$ sensitivity or no temperature sensitivity, a finding consistent with the fall off in decay seen in the incubations (Fig. 2). Surprisingly, after

controlling for temperature response, each of the four moisture functions had similar AIC scores, with no single moisture model performing appreciably better than the others (Fig 6).

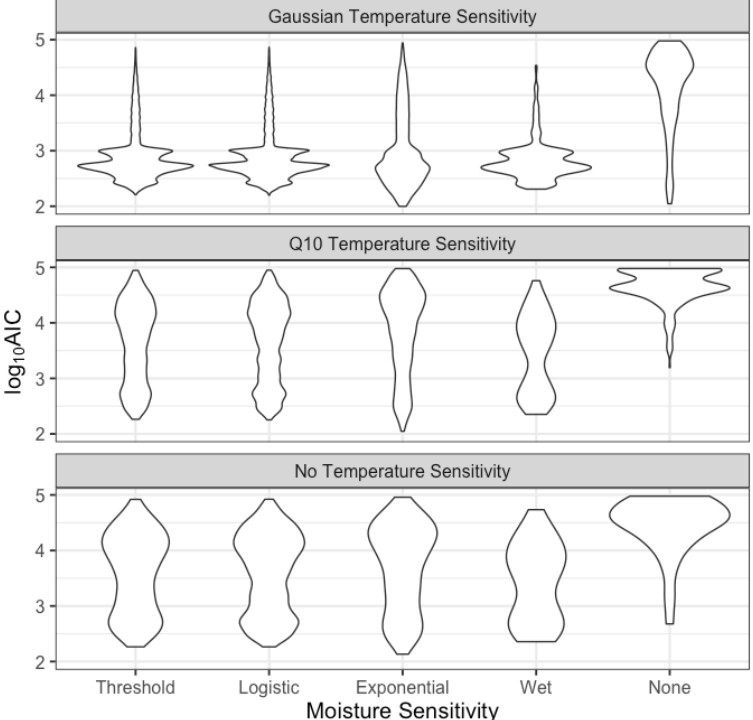

**Figure 6: Model performance ($\log_{10}$AIC scores) for each model combination of temperature and moisture sensitivities. Each observation represents one parameter combination after constraining them as described in Table 2. Lower scores**

**denote better fits. This only shows models constrained using realistic parameter estimates (n = 895,230).**

Including temperature sensitivity alone (without NRM) did not improve model fit as well as modeling only NRM sensitivity (without temperature). All of the NRM-only models (Fig 6, bottom row) had better fits than did temperature-only models (Fig 6, right column), though each showed a wide range depending on the specific parameter combinations. In fact,

an unconstrained model with $Q_{10}$ temperature sensitivity but no moisture sensitivity converged on an optimal $Q_{10}$ value <1,





indicating a negative temperature dependence of litter decomposition (Fig. A6), the opposite of what we observed in the temperature incubations.

When we compared one of the best models that included temperature and NRM sensitivity (specifically, a Gaussian temperature function and an exponential moisture function) to a simple decay model that had no temperature or NRM
sensitivity but varied effective decay rate among sites (Equation 1), we found that the temperature and NRM model performed better (Fig 7). The Gaussian-exponential model had lower AIC scores and the slope of the observed vs. predicted values was closer to 1, yielding more realistic mass loss predictions (0.85 for Gaussian-exponential model, 0.71 for simple decay model).

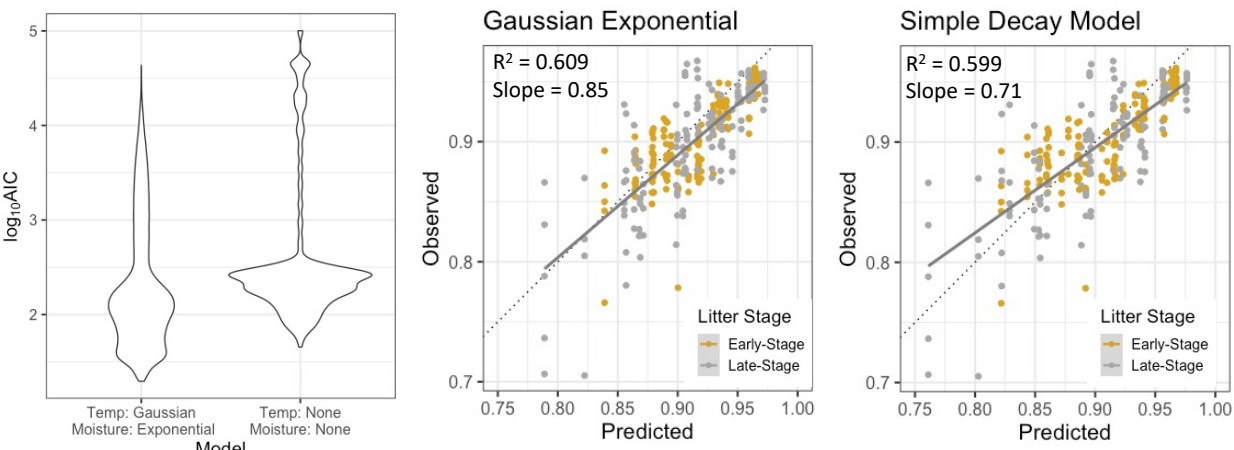

**Figure 7: (Left panel) Fit of the model using Gaussian temperature sensitivity and exponential moisture sensitivity**
**versus a simple exponential decay model (without temperature or NRM sensitivity) in which $K_{eff}$ is allowed to vary independently for each site. The simple decay model depicted here differs from the one in Fig 6 because this one uses $K_{eff}$ and is not constrained to the same set of parameters from there. (Right two panels) Model predictions for the best version of the Gaussian-Exponential model versus the simple decay model with site-specific $K_{eff}$ values. Solid lines are the best fit lines and dotted lines are the ideal 1:1 line.**


## 4 Discussion

### 4.1 Model performance

Decomposition is a crucial component of Earth system models and NRM is an important moisture source in arid and mesic grasslands worldwide. In a first attempt at modeling NRM-driven decomposition, Evans et al. (2020) compared litter
decay rates in a hyperarid and a mesic grassland, showing that decay rates are faster when NRM is more frequent. We build on this work by demonstrating a scalable quantification of the relationship between NRM, temperature, and litter decay rates. Doing so is an important step to improving Earth system models, which must be validated with field measurements made under





realistic conditions (Bonan et al., 2013). Using a 30-month, multi-site field experiment, we show that explicitly accounting for both temperature and non-rainfall moisture sensitivity improved a litter decay model in an NRM-affected system.

While incorporating either NRM sensitivity or temperature dependence improved model performance, it was the inclusion of both that led to the largest improvement. Decomposition's temperature sensitivity often depends on moisture conditions (Petraglia et al., 2019). For example, in soils, temperature typically increases decay rates when moisture is abundant, but higher temperatures can dry out soils, slowing decomposition (Bear et al., 2014). Similarly, in our system, NRM increases litter moisture content (Jacobson et al., 2015), but fog and dew only form at cooler temperatures, when decomposition

is slower; once temperatures get high enough (in this case, above 20°C; Fig. A3), wet conditions soon cease, making the positive effects of temperature moot. We find that this nuance about NRM gives rise to unrealistic predictions when models include only one type of sensitivity but not the other. For example, in our unconstrained model run, a model with only temperature dependence, but no NRM sensitivity, converged on a $Q_{10}$ temperature sensitivity <1, indicating negative temperature dependence (Fig. A6), even though incubation data clearly show a positive relationship across the range of

conditions tillers experience in the field (Fig 2). This shows that both temperature and NRM sensitivity were needed to realistically capture litter decay dynamics under NRM conditions, lest one mask the effects of the other, yielding unrealistic results.

The choice of temperature sensitivity function is often very important in modeling biological processes and can lead to quite different predictions (Low-Décarie et al., 2017). We found that model performance was better using a Gaussian rather

than a $Q_{10}$ temperature sensitivity function. Surprisingly, we found that the different NRM sensitivity functions, including both continuous and threshold functions, described litter decay dynamics similarly well. While the threshold, logistic, and wetness moisture sensitivity functions share a general form in which decomposition rates increase substantially above a specific relative humidity value, the exponential function simulates gradually increasing decay rates at different relative humidity values. In this sense, the exponential function more accurately mimics the moisture absorption curves seen in field

and lab studies (Dirks et al., 2010; Evans et al., 2020; Tschinkel, 1973). Despite these differences, however, each of these functions led to similar model performance. This suggests that, while explicitly including sensitivity to NRM is important, the specific manner in which moisture is represented in the model may be less important. NRM-explicit litter decay models in the future may be able to represent NRM with fewer parameters by adopting a simple threshold approach, eliminating the need to parameterize multiple moisture components. Since relative humidity is a standard meteorological measure (unlike leaf

wetness), future models should be able to use existing data sources to incorporate NRM, eliminating the need to collect additional data with specialized instrumentation (Evans et al., 2020).

## 4.2 Litter properties

Surprisingly, early- and late-stage litter had similar relative humidity thresholds for decomposition even though older litter tends to absorb more water during fog and dew events (Logan et al., 2022). In the absence of rain, litter moisture content

rarely reaches biologically significant levels until relative humidity reaches at least 70-80% (Dirks et al., 2010; Evans et al.,



2020; Tschinkel, 1973), but this depends on several factors including the permeability of the litter to water, the amount of time it spends in humid conditions, and the decomposer community's sensitivity to moisture (Tschinkel, 1973; Logan et al., 2021, 2022). While the simple threshold-based moisture function performed very well in this study, future studies will likely need to parameterize the moisture threshold to fit the dominant litter type in their locales.

Despite converging on the same parameter values, model fits were much better for late-stage litter than for early-stage litter (Fig. A5). This could reflect the fact that the larger fungal communities on late-stage tillers enable them to respond to moisture more strongly than early-stage tillers, which do not have a large enough decomposer community to have a strong biological response to NRM yet. This is consistent with the results from our simple decay model (without explicit temperature and moisture sensitivity), which showed that effective litter decay rates for late-stage tillers were 3.3 times more sensitive to
changes in NRM frequency than were early-stage tillers (Fig 4). Since the major differences between the early- and late-stage tillers we used in this study are their degree of prior fungal colonization and their ability to absorb water, this reinforces the importance of fungal communities as mediators of decomposition's response to NRM (Logan et al., 2021) and suggests that plant litter properties related to moisture absorption may influence NRM-sensitivity (Logan et al., 2022). Examining whether these properties have the same influence on NRM-driven decay of other plant species may increase the generalizability of the
response functions we present here.

### 4.3 Incorporating into existing Earth system models

Developing models that realistically predict carbon turnover is a multi-step process that requires determining a model structure, parameterizing, and accounting for external forcings (Luo et al., 2015). Our goal was to compare several potential structures for modeling NRM-driven litter decomposition, but fully incorporating NRM sensitivity into existing Earth system
models will require additional work. This includes identifying the appropriate temporal resolution at which to model NRM events. The timesteps used by Earth system models have shortened considerably over the last two decades, to the point where processes that were once represented monthly are now modeled on hourly timescales or less (Sokolov et al., 2018; Bolker et al., 1998; Bonan et al., 2013). We used hourly averages of minute data to describe decomposition rates, but do not yet know what temporal resolution is necessary to fully capture NRM events. Future studies can compare estimates using minute data
(that have the benefit of capturing the wetting and drying dynamics of litter at the start and end of NRM events) to daily timescales, that may estimate NRM-driven decomposition from daily mean relative humidity. In the case of longer (daily) timescales, temperature dependence may be best determined using the minimum daily temperature instead of mean temperature, since minimum temperatures are likely to occur at night when NRM is most common. Of course, these methods will require additional testing, but since our models were relatively insensitive to the specific nuances of how NRM was
modeled, any of several approaches may be appropriate depending on the structure of the decomposition model in use.

We used relative humidity and leaf wetness sensor data to parameterize our moisture sensitivity functions but other methods of modeling moisture may work as well. Many ecosystem models treat soil water content (which regulates soil organic matter decomposition) as related to the ratio of rain to evapotranspiration (Necpálová et al., 2015). If NRM-driven



decomposition can be captured by proxies constructed from evaporation, minimum temperature, and other values already
included in carbon sub-models, it may be easier to incorporate this novel process into existing modeling approaches.
Fortunately, relative humidity is measured at meteorological stations worldwide and extensive data are available. Even in
regions with data gaps, methods exist to estimate relative humidity from temperature datasets (Gunawardhana et al., 2017) and
these can be incorporated into Earth system models to include NRM sensitivity without the need to collect additional data.

While our study focused exclusively on aboveground litter decay, NRM may have other effects on decomposition
later in the decay process as well. NRM-driven decomposition removes carbon from the system before it reaches the soil
surface, decreasing inputs to belowground pools. Additionally, NRM may accelerate belowground decomposition rates once
litter is incorporated into the soil by promoting the development of larger (and specialized) microbial communities early in the
decay process (Logan et al., 2021; Jacobson et al., 2015). Such soil-litter mixing often increases litter decomposition in dryland
systems (Barnes et al., 2015, 2012; Hewins et al., 2013). Even more broadly, there are other processes for which models
ignore the role of NRM that affect carbon cycling, like stimulating plant growth, and suppressing wildfires (Weathers, 1999;
Emery et al., 2018). To improve our understanding of NRM-driven decomposition, studies can test the role of NRM-driven
decomposition on both aboveground and belowground litter to identify how NRM affects linkages between these two pools.

### 4.4 Limitations

Since our goal was to present a first attempt at incorporating NRM into litter decay models in an NRM-dominated
ecosystem, we had to make several simplifications that likely underestimated litter decay rates. First, we only looked at coarse
tillers, not leafy material. In lab incubations, Jacobson et al. (2015) found that at high humidity, the water content of *S.
sabulicola* tillers (like those we used) increased slowly, reaching only 10.5% after 2 hours, with no detectable $CO_2$ flux. In
contrast, fine leaves reached a moisture content of 30.3% and exhibited a flux rate of 0.99 µg $CO_2$-C g litter$^{-1}$ min$^{-1}$ after 2
hours. In a field study, Evans et al. (2020) showed that gravimetric moisture content of *S. sabulicola* tillers could reach 0.35
g $H_2O$ g litter$^{-1}$ while leafy material could absorb as much as 1.45 g $H_2O$ g litter$^{-1}$ during NRM events, resulting in considerably
higher respiration rates for leaves. Similarly, windblown detritus (litter that has become physically disconnected from the
plant) makes up a considerable proportion of total litter mass in the Namib (Seely and Louw, 1980) and can absorb substantial
water under humid conditions (Tschinkel, 1973). As a result, actual rates of NRM-driven decomposition across the whole
landscape are likely higher than what we report here.

Secondly, we focused only on the meteorological drivers of litter decomposition, though others factors play important
roles as well. Photodegradation (Austin and Vivanco, 2006; King et al., 2012), macrodetritivore activity (Louw and Seely,
1982), and soil-litter mixing (Hewins et al., 2013; Lee et al., 2014) are all important drivers of litter decomposition in drylands.
Since our goal was to quantify the relationship between NRM and litter turnover, we focused solely on NRM, but future studies
can build on this work by combining our approach with other existing models. For instance, photodegradation can interact
with NRM to accelerate carbon turnover, especially of standing litter (Wang et al., 2017; Logan et al., 2022), and accounting
for photodegradation improves litter decay models (Chen et al., 2016; Adair et al., 2017). Combining these other mechanisms





with the relative humidity-based litter decay model we present here may reveal additional interactions that can be validated by field studies. The fact that we were able to describe a large degree of litter decomposition by using a simple relative humidity-based and temperature-based model, however, demonstrates that NRM plays an important role in the litter decay process across a wide range of environmental conditions.

## 4.5 Conclusions

We show that the frequency of non-rainfall moisture is a major predictor of litter decomposition, and for the first time, used data from a multi-site field study to develop temperature and NRM sensitivity functions for a litter decay model. Temperature and moisture regimes are changing as a result of anthropogenic climate change (Byrne and O'Gorman, 2016) and our ability to predict how ecosystems respond depends, in part, on how well we can link biogeochemical cycles to their environmental drivers. NRM and rainfall are often controlled by different climatic drivers and may therefore respond differently under future climate change (Haensler et al., 2011; Dai, 2013; Forthun et al., 2006). By modeling the contribution of NRM to decomposition, in addition to that of rainfall, we can better predict how drylands will respond to changing moisture regimes, increasing our ability to manage these globally important systems.





# 5 Appendix

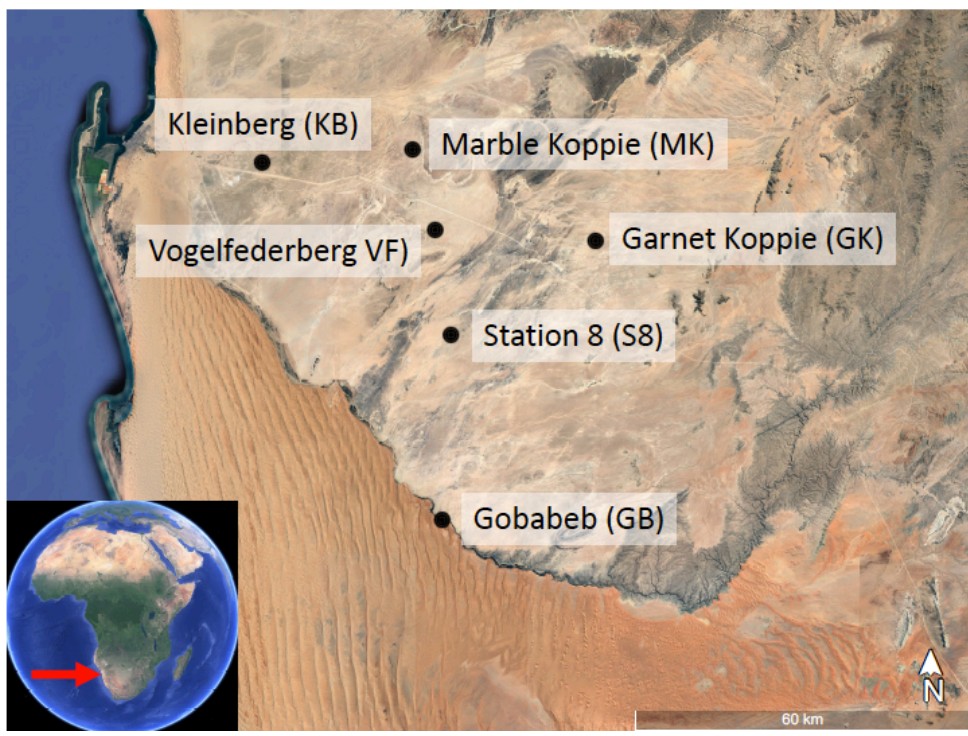

**Figure A1: Location of the six FogNet sites used in this study. All samples were collected from dunes of the Namib Sand Sea at Gobabeb. Background image provided by © Google Earth.**



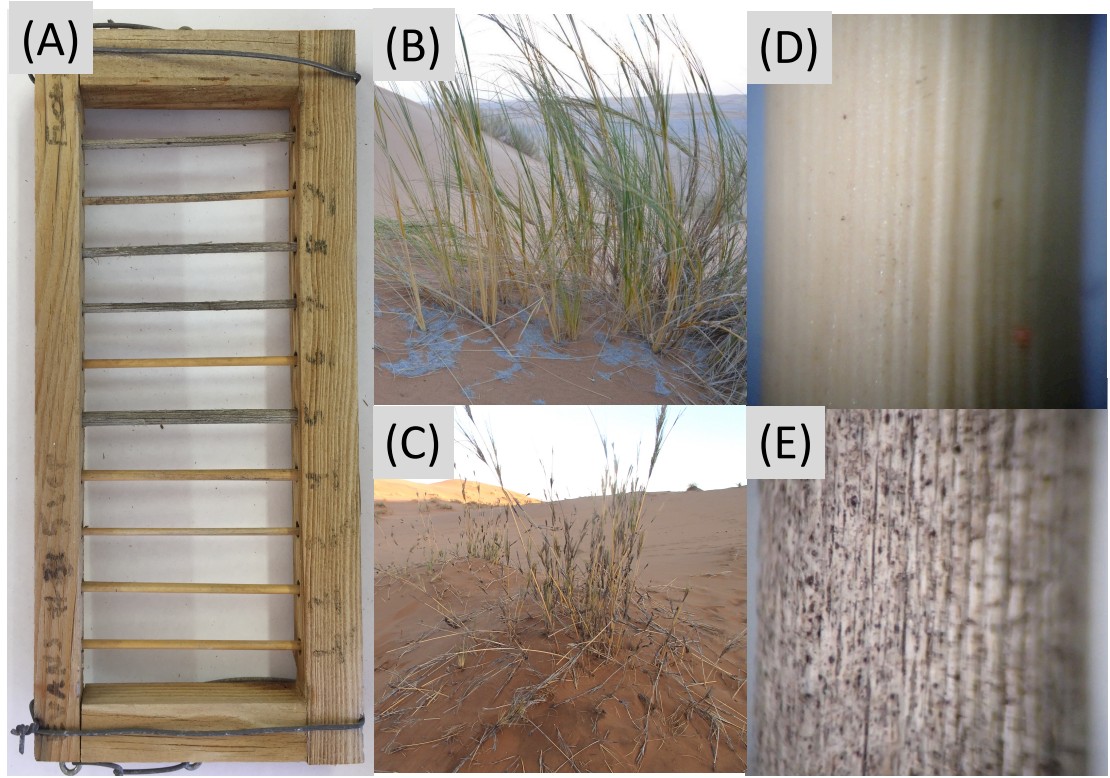

**Figure A2: (A) Example of a litter rack used instead of litter bag. The "rungs" of the "ladders" are _Stipagrostis sabulicola_ stems ~ 0.5 cm in diameter and 9 cm long; (B) Living _S. sabulicola_ hummock growing in the dunes; (C) Dead**
**_S. sabulicola_ tillers like those used in this study; (D) Close up image of a recently senesced (early-stage) tiller with inact cuticles and little fungal growth; (E) Close up image of a late-stage tiller with cracked cuticle surface and substantial colonization by dark pigmented fungi.**





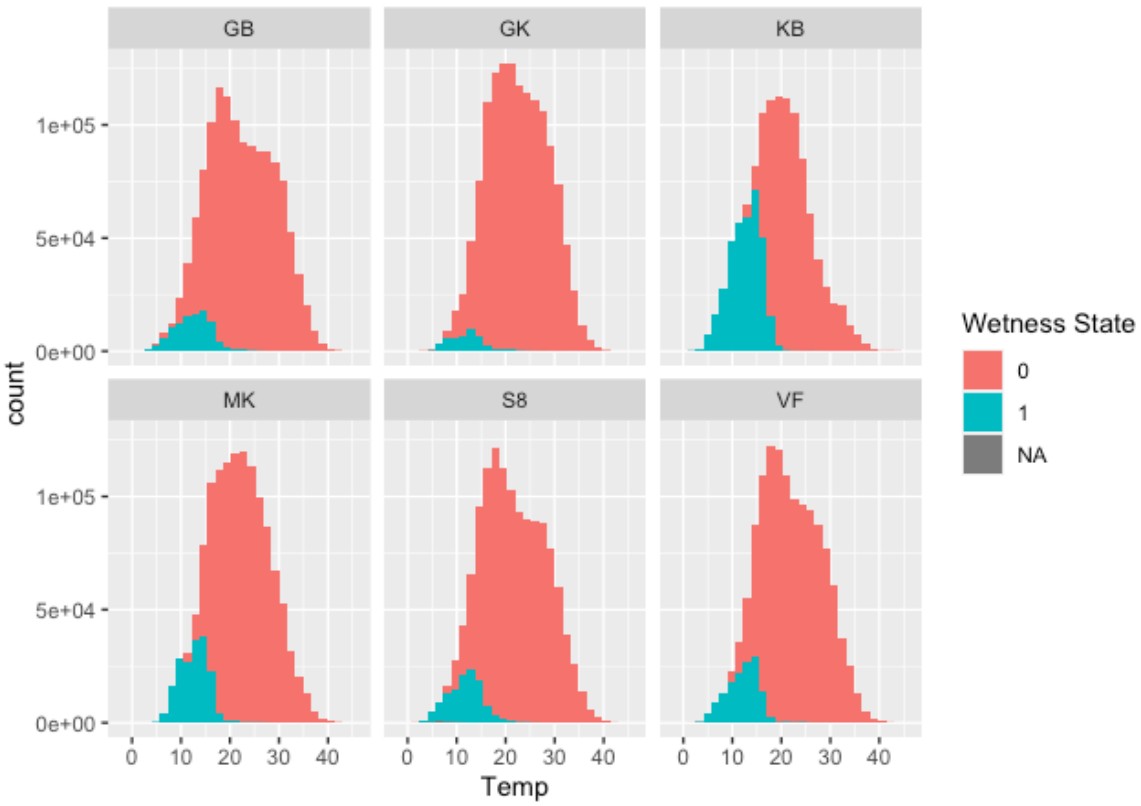

**Figure A3: Frequency distributions of temperature when wet (blue) and dry (red) at the six sites during the study.**


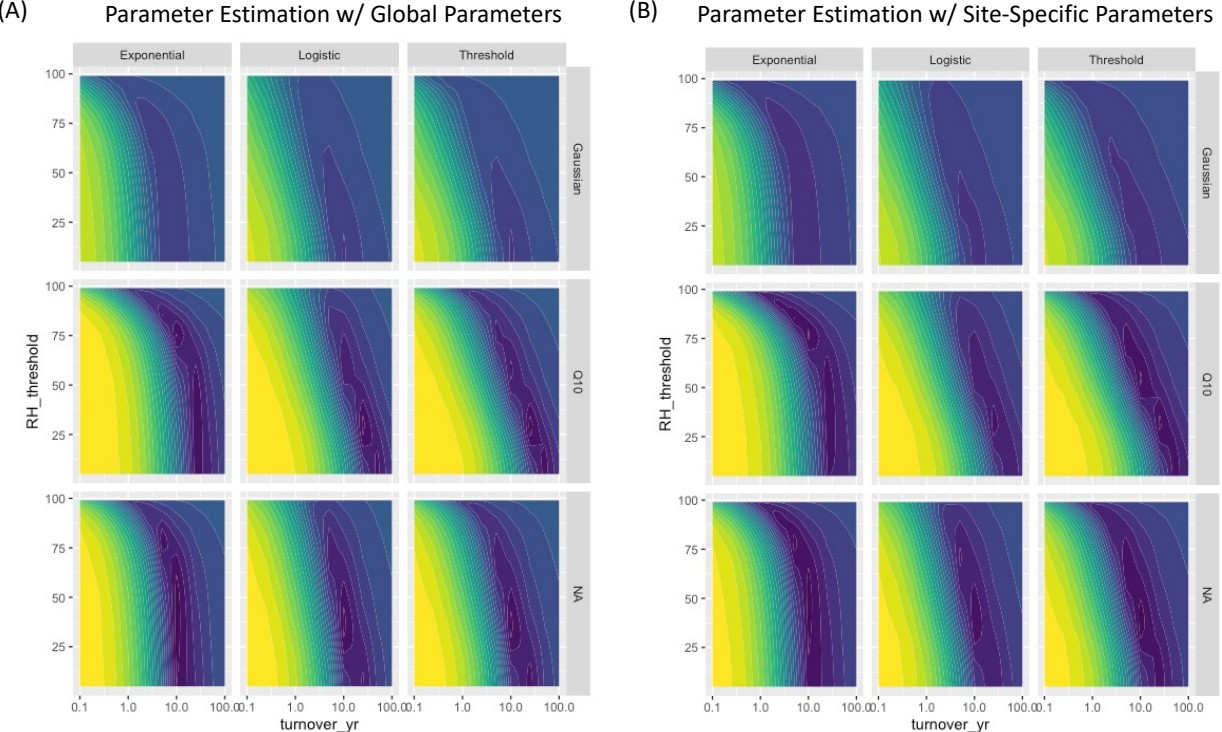

**Figure A4: Parameter fits for the humidity-based moisture models using (A) global parameters that were fitted across sites, and (B) site-specific parameters. Colors represent AIC scores with purple denoting lower values and yellow denoting higher values. The left panel is identical to Fig 3 in the main text.**



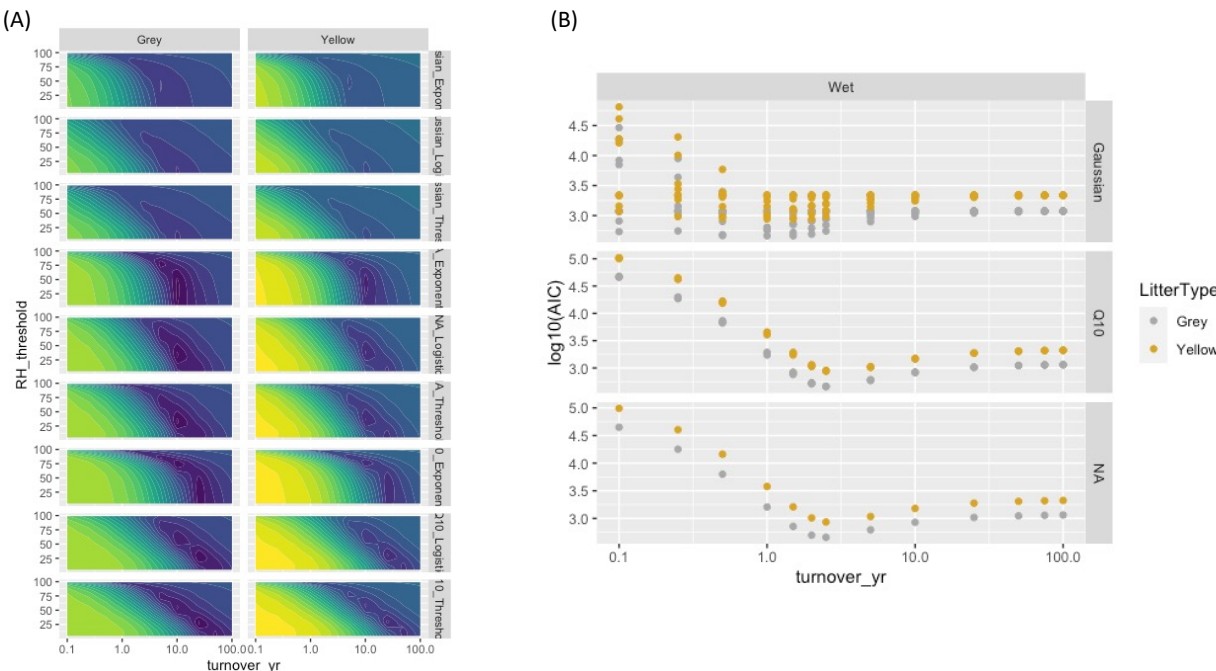

**Figure A5: (A) Parameter fits for the humidity-based models for (left) late-stage litter and (right) early-stage litter. Colors represent AIC scores with purple denoting lower values and yellow denoting higher values. (B) Model fits for the wetness-based models, color-coded by litter stage (this is identical to the right panel of Fig 5, but color coded to show differences in litter stage).**

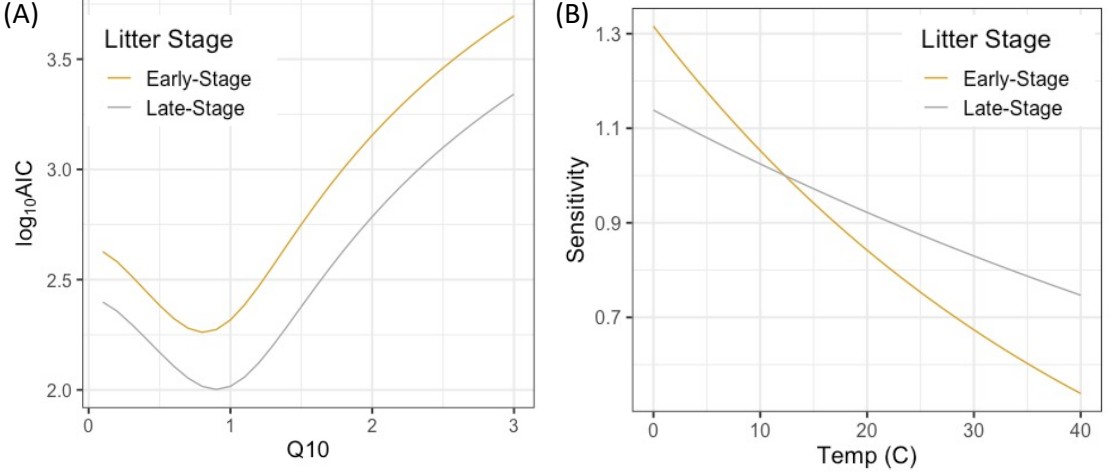

**Figure A6. (A) Parameter estimation plot of $Q_{10}$ coefficient for an $Q_{10}$-only model run (i.e. with no NRM sensitivity) showing model fit is best for $Q_{10}$ values below 1. (B) Estimated $Q_{10}$ sensitivity curves based on optimal value determined from panel A.**





## 6 Code and data availability

Data and code used in this paper are available as an R Markdown file at https://github.com/loganja3/NRM-Gradient-Project

## 7 Author contributions

JRL, SEE, PJJ, and KMJ conceived of the study. JRL designed the study and conducted fieldwork. JRL and KTB developed the model code and performed the simulations. PJJ and KMJ performed the lab incubations. RV collected and processed meteorological data. JRL conducted data analysis and statistics and prepared the manuscript with contributions from all co-
authors.

## 8 Conflict of interest

The authors declare that they have no conflict of interest.

## 9 Acknowledgements

We would like to thank the staff at the Gobabeb Namib Research Institute, the Namibian Ministry of Environment and Tourism,
the National Botanical Research Institute, and the National Commission on Research, Science and Technology (permit number RPIV000102017) for their permission and support for this study. Special thanks to Martin Handjaba at Gobabeb for his fieldwork in support of this project and to Lukas Mendel for assistance with lab incubations. Funding was provided by the U.S. National Science Foundation's Graduate Research Fellowship Program, the W.K. Kellogg Biological Station at Michigan State University, Grinnell College, and the taxpayers of the United States and Michigan. The Southern African Science Service
Centre for Climate Change and Adaptive Land Management provided the initial funding to establish the FogNet weather network and it is currently maintained by funding from the University of Basel. This work was also supported in part through computational resources and services provided by the Institute for Cyber-Enabled Research at Michigan State University. Thank you to the participants at the 2014 FogLife Colloquium, in particular Mary Seely and Theo Wassenaar, for their contributions in helping generate ideas for this project from the start. Finally, thank you to the Evans Lab, the KBS Writing
Group and two reviewers for their feedback on earlier versions of this manuscript. This paper is KBS contribution number 2303.

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
