# Peer review of "Accounting for non-rainfall moisture and temperature improves litter decay model performance in a fog-dominated dryland system"

_Biogeosciences, 2022_

## Author Response (AR1)

We would like to thank the reviewer for their comments and note that the original review is in *italic* with our reply following.

*The paper addresses effects of non-rainfall moisture on rates of litter decomposition, which is an important pathway of litter degradation in dryland system. Solely considering sparse rainfall is inadequate because it does not account for moisture delivery through air saturated water in form of dew and fog. The paper details a 30 months desert litter decomposition experiment. Along periodical litter harvest, environmental data (moisture, temperature has been measured). Analysis with a broad palette of temperature and moisture functions shows that there is an important impact of soil moisture.*

*I like this paper a lot, it effectively combines a thought-through experimental setup (moisture delivery gradient, quite long-term) with appropriate modeling experiments. I like the use of different functions to reduce effects of the functional forms (for example in that an optimum temperature maximum may impact temperature effects at higher temperature). The paper also points to critical limits of Earth system models.*

*There are just a few minor comments that came to my mind when reading the paper.*

*I am wondering to what degree this type of experiment could also help with litter decomposition in more moist system. Typically, models would consider soil moisture that does not reflect the moisture in the surface litter. Could this experiment inform this too?*

This is an excellent point and we thank the reviewer for these comments. Yes, our study was conducted at a site with extremely little rainfall and we are therefore not able to directly extrapolate the effect of NRM on mass loss in more mesic sites. However, NRM's role in litter decay has been observed in a wide range of ecosystems including Mediterranean shrublands (Gliksman, 2018; Dirks et al., 2010), salt-marshes (Newell et al., 1985), hyperarid deserts (Logan et al., 2021), and temperate steppes (Wang et al., 2017). One study found that NRM played a substantial role even a mesic prairie with mean annual precipitation of 897 mm (Evans et al., 2020), suggesting that NRM is important even when rainfall is relatively frequent. Our contribution in this paper is therefore not demonstrating the importance of NRM to litter decomposition in general, but showing that the frequency of NRM events strongly predicts litter mass loss across a wide range of moisture conditions and that this can be easily modeled using relative humidity data. Although this study was conducted at the dry end of an aridity gradient, it still represented an eight-fold magnitude of NRM frequency, showing that NRM can be easily incorporated into litter decay models. Explicitly incorporating NRM into models in mesic systems, where rainfall plays a greater role, will likely require including both rainfall and NRM-sensitivity functions to identify the relative role of each as rainfall increases.

We have elaborated on this point by including a paragraph in the discussion to discuss these points and believe that this strengthens the paper. Thank you again for your questions and suggestions!

We would like to thank the reviewer for their comments and note that the original review is in *italic* with our reply following.

*This manuscript describes a study that quantified the effects of non-rainfall moisture on litter decomposition across a fog gradient in the Namib desert. Non-rainfall moisture (relative humidity and dew) is thought to be important source of moisture in very dry ecosystems, yet few studies have quantified the effects on litter decomposition. The manuscript describes the results of a multi-year litter mass loss study and tests different temperature and moisture relationships in a simple decay model. They show that including non-rainfall moisture improves the decay model performance.*

*I enjoyed reading this manuscript. It is very well written and I really appreciate the author's efforts to submit a polished document. I did not see any grammatical or technical mistakes. The study is interesting and provides a novel model that could be tested in other ecosystems. The methods use were appropriate and sound to my knowledge.*

*I do wonder how applicable is this model/study to other ecosystems? The Namib desert is extreme, would this non-rainfall moisture effect be as prominent in a less extreme but still foggy ecosystem? Is this effect big enough to detect in ecosystems with more decomposition?*

We thank the reviewer for this comment.

NRM is an important driver of litter decomposition in both arid and mesic systems (Evans 2020) and while litter decomposition typically happens faster belowground than on the surface, surface litter decomposition is an increasingly recognized part of dryland carbon cycling. Standing litter often decomposes faster than litter lying at the soil surface (Liu et al. 2015 Plant Soil; Gliksman et al. 2018 Plant Soil) and represents an important and, until recently, overlooked source of carbon turnover in drylands (Wang et al. 2017 Functional Ecology). While we did not look at litter at the soil surface, surface litter absorbs atmospheric moisture (Tschinkel 1973 Madoqua) and may respond similarly to NRM, though to date, no models we know of have looked at this across a range of NRM conditions.

Our study demonstrates that NRM can predict standing litter decay rates and since surface litter decay rates are also strongly affected by relative humidity (Wang et al. 2017), we would expect that a similar approach could be used to model NRM's effect on surface-litter decomposition rates. This would of course require field studies to validate and parameterize an NRM-driven surface-litter decay model. We plan to include a few sentences elaborating on this point in our revision.

*Difference between older senesced vs. new senesced litter: I think that difference could be a bit more leveraged – the main difference is that older litter may be a bit more populated with decomposers – could this be a way to address the importance of colonization, and how we need to think about that in decomposition? Can that difference be a bit more discussed? What was the motivation behind this doubling of the data – is there an objective going with that?*

We recognize that using litter at different stages is a little unusual and we thank the reviewer for the opportunity to elaborate on our choice here.

Pre-colonization is a very important step in standing-litter decomposition since it can "prime" litter to be more ready to degrade once it reaches the soil surface. Previous work in the Namib (conducted near the middle of the moisture gradient used in this study) showed that pre-colonized litter degrades considerably faster than recently senesced litter (Logan et al. 2021). When we established this gradient-based study, we did not know how much more quickly or slowly litter would decompose when moved to drier or wetter conditions. We deployed litter at two different initial stages so we could compare decomposition rates among the sites, even if it progressed faster or slower than expected.

Indeed, our data showed that early- and late-stage litter responded differently to moisture conditions. At the two driest sites, early-stage litter decomposed faster than late-stage litter (Figure 3; Garnet Koppie and Gobabeb), possibly reflecting the contribution of photodegradation early in the decomposition process when water is limited (Logan et al. 2022 Functional Ecology). By contrast, at the wetter sites (especially the wettest site, Kleinberg), late-stage litter decomposed faster, likely because saprophytes that had pre-colonized the litter could make use of the readily available water at these sites and support greater mass loss. By using both litter types, we could identify some of these patterns that might otherwise have taken many more years for the study to progress (i.e. if we had waited for the recently-senesced tillers to decompose completely).

To make better use of this part of the experiment, we plan to include a few sentences more explicitly discussing these points in the revision.

*Detailed/Technical/Minor comments*

*L85: While technically correct, many today assume with the Bayesian approach the inference of a parameter uncertainty range – which is here not provided (and not necessary). Perhaps use Bayesian/Maximum Likelihood approach?*

Thank you for pointing out that this could be clearer. Due to the relatively simple nature of this model and restricted parameter range we used an iterative Monte Carlo approach with an AIC fit function and prior parameter distributions given in Table 2. We propose changing the wording here from "Bayesian" approach to "Bayesian - Monte Carlo" to better reflect this.

*Methods: What function was used to optimize – minimize R2 and standard normal distribution or any other functional form (Gamma etc.) This may be in the provided code (which admittedly I have not reviewed), but it would be good to mention it explicitly in the paper.*

Thank you for pointing this out; this was not clear in the manuscript. We did not optimize the function but instead used a brute-force approach where we randomized parameter inputs to represent conditions seen in the field (Table 2) and then selected the model-parameter combinations with the lowest AIC scores. Our parameter space was small enough that we were able to run the model with a full range of parameter combinations of interest without having to optimize the model directly. We are making this clearer in the methods by explicitly stating this in Section 2.4 "Constraining parameter space" to clear up this point for future readers.

*L138: strictly speaking, this integral of the linear decomposition model is only correct when g(t) and h(t) are constants between beginning and end of time interval.*

Thank you for pointing this out! We agree and missed this in our earlier reading. Since the equation we specifically used in our code is the discretized form (Eqn 4) of the instantaneous litter decay rate (Eqn 2), the integral is not essential right here and we plan to remove this equation to focus on the ones we actually used in the model.

*Figure 1 /Table 1 and related method text: I really appreciate the authors effort to evaluate different functional forms of g(t) and h(t)*

We thank the author for their comment.

*L 273: Please add unit (yr-1) to the effective decay rate numbers*

Thank you for pointing this out. We have made this change.

*Reporting of AIC scores: I think for many of the readers it would be helpful to mention in at least one of the figure captions that you aim to minimize the AIC scores*

This is a very good point. We have added this to each caption to make this point clearer to readers.

*Figure 6: is it clear that the figure represent some sort of probability density distributions of AIC scores?*

We have elaborated in the caption for Figure 6 to explain that it is showing Frequency distributions of AIC scores for the different model-parameter combinations.

*Figure 5A: Please change the Litter Type legend to "Early Stage" and "Late Stage".*

We have made this change in both figure panels.

*The conclusions drawn are carefully laid out, based on carefully laid out evidence, without overstating. I like that!*

Thank you!